# Comparative Study on Photocatalytic Performance of TiO₂ Doped with Different Amino Acids in Degradation of Antibiotics

Hadis Zangeneh [1], Seyyed Alireza Mousavi [1,2,*], Parisa Eskandari [3], Ehsan Amarloo [4], Javad Farghelitiyan [5] and Sahar Mohammadi [6]

1. Department of Environmental Health Engineering, School of Public Health, Research Center for Environmental Determinants of Health (RCEDH), Kermanshah University of Medical Sciences, Kermanshah 51351, Iran
2. Social Development and Health Promotion Research Center, Kermanshah University of Medical Sciences, Kermanshah 51351, Iran
3. School of Chemical Engineering, The University of New South Wales, Sydney, NSW 2052, Australia
4. Department of Chemical Engineering, Sharif University of Technology, Tehran 11155, Iran
5. Department of Civil Engineering, Technical and Vocational University, Isfahan 73441, Iran
6. School of Chemical and Petroleum Engineering, Shiraz University, Shiraz 16511, Iran
* Correspondence: sar.mousavi@kums.ac.ir

**Abstract:** In this study, three different reusable photocatalysts containing different amino acids as a source of non-metals, including L-Arginine, L-Proline, and L-Methionine, have been synthesized for the first time. Using a kinetic study and degradation efficiency test, these visible driven photocatalysts were investigated for their photocatalytic activity in removing antibiotics, including metronidazole (MNZ) and cephalexin (CEX). The morphology, structure and optical properties of the fabricated catalysts were characterized by X-ray Powder Diffraction (XRD), Field Emission Scanning Electron Microscopy (FESEM), Energy Dispersive Spectrometry (EDS)/mapping, Fourier-Transform Infrared Spectroscopy (FTIR), Photoluminescence Spectroscopy (PL) and UV-Vis Diffuse Reflectance Spectroscopy (DRS) analyses. Based on the results of the PL analysis, it was confirmed that doping TiO₂ with amino acids containing C, N, and S inhibited the recombination of induced electrons and holes. Among the three catalysts, L-Arginine-TiO₂ demonstrated the highest photocatalytic activity for antibiotic degradation, followed by L-Proline-TiO₂. According to the response surface methodology (RSM), the optimum operating conditions were a concentration of 50 mg/L MNZ, pH = 4, and catalyst concentration = 1.5 g/L under 90 min of irradiation time. At this condition, 99.9% of MNZ and 81% of TOC were removed. In addition, 97.2% of CEX and 75% TOC were eliminated at the optimum conditions of 1g/L catalyst concentration, 50 mg/L CEX concentration, at neutral pH, and after 120 min irradiation. L-Arginine (1 wt.%)-TiO₂ was tested for stability and reusability, and it showed that after five cycles, 10% of its performance had been lost. The role of reactive species in photocatalysis was identified and •OH had the most significant impacts on MNZ and CEX photodegradation. Antibiotic degradation efficiency was adversely affected by the presence of anions and humic acid, but this reduction was not significant for inorganic anions, as only 13% of degradation was lost.

**Keywords:** L-Proline-TiO₂; L-Arginine-TiO₂; L-Methionine-TiO₂; photocatalyst; antibiotic; scavengers

## 1. Introduction

Residues of pharmaceutical compounds, especially antibiotics, are considered emerging environmental contaminants. These contaminants can be found in water resources and wastewater around the world [1] as a result of their extensive use in human and livestock medicine [2]. In the body, antibiotics are poorly absorbed and approximately 50–90% will be excreted into the environment; since total antibiotic consumption worldwide is 100,000 to

200,000 tons, it is necessary to remove them from effluents. [3]. The presence of antibiotics such as metronidazole and cephalexin in the environment can adversely affect flora, fauna, and humans [4] even at low concentrations through ecotoxicity [5], potential mutagenicity, carcinogenicity, chronic toxicity, and microbial resistance, making them a global threat to health [6]. Due to the chemical structures and bio-recalcitrant nature of antibiotics, conventional methods cannot effectively remove them from polluted wastewater. Therefore, advanced alternatives are needed [7,8].

For the treatment of wastewater containing antibiotics, semiconductor photocatalysts have gained considerable attention as an effective and promising method. [9,10]. In this method, photocatalytic oxidation reactions generate reactive species under light exposure, such as hydroxyl radicals [11,12]. The advantages of photocatalysis are that it is green, energy-efficient [13], and has lower costs; it is also easy to obtain, non-toxic, stable, and reusable [14]. Consequently, it is a feasible and sustainable method for photodegrading antibiotics [15].

As a semiconductor, $TiO_2$ has a high potential for photocatalytic degradation due to its excellent optical properties, chemical resistance [16], strong oxidizing power [17], and low cost and toxicity [18]. Under visible light illumination, $TiO_2$ cannot function effectively due to its large band gap and rapid electron–hole recombination rate. As a result, it does not meet the requirements for practical application [19,20]. In order to enhance $TiO_2$'s photocatalytic activity, it needs to be modified [21], which is the main focus of this research. Despite N-doped $TiO_2$'s potential for enhancing photocatalytic activity, the materials are not very stable under visible light exposure due to the low amount of nitrogen dopant [22]. Furthermore, C-$TiO_2$ can have positive effects in facilitating electron/hole transfer from $TiO_2$ to pollutants. In addition, carbon acts as a trap for photo-generated electrons, enhancing C-$TiO_2$'s visible-light photocatalytic activity [22,23]. Recent studies showed that co-doping and tri-doping of $TiO_2$ significantly enhanced the photocatalytic performance of $TiO_2$ in the visible light region due to its synergistic impacts [24]. The study of Xiao et al. found that co-doped-$TiO_2$ particles removed tetracycline at a rate of about 86.3% after 30 min visible light irradiation [23]. Wan et al. showed that phenol removal reached 43% using bare $TiO_2$ under visible light irradiation, whereas 90% of this pollutant was removed by C,N,S tri-doped-$TiO_2$ as a photocatalyst [25]. However, these studies did not assess the simultaneous effects of processing factors as well as their interactions with the photodegradation process. In these studies, using a one-variable-at-a-time technique was considered for the design and assessment of experiments which has some disadvantages of overlooking the interaction between processing factors and increasing the number of experiments.

As a mathematical and statistical method, the response surface methodology (RSM) can help to determine the effects and optimum amounts of processing factors. [26,27]. The mathematical modeling of the photocatalytic removal of antibiotics using RSM has only been investigated in a few recent studies. In this regard, the aim of this study is to synthesize different types of multi-doped $TiO_2$ photocatalysts, L-Proline-$TiO_2$, L-Arginine-$TiO_2$ and L-Methionine-$TiO_2$ and to provide their characterization to compare their photocatalytic performance in MNZ and CEX removal. In addition, the effect of processing parameters, initial MNZ or CEX amount, pH, catalyst dosage, and irradiation time on degradation efficiency is analyzed. Determination of optimization conditions and the reusability of the nanoparticles is also carried out.

## 2. Experimental

### 2.1. Synthesis of Amino Acid-$TiO_2$

The synthesis of three types of photocatalysts including two types of C/N/$TiO_2$ (L-Proline-$TiO_2$ and L-Arginine-$TiO_2$) and C/N/S/$TiO_2$ (L-Methionine-$TiO_2$) was performed based on the sol-gel method [28]. In total, 5 mL of TBOT ($C_{16}H_{36}O_4Ti$, purity 99%, Merck, Darmstadt, Germany) was dissolved in 10 mL ethanol (Merck, purity 99%) followed by adding 3.4 mL of acetic acid (purity 99%, Merck) and the solution was mixed and dispersed

for 20 min in an ultrasonic bath. Various weight proportions (0.5, 1, 1.5, 2, 2.5 wt.%) of L-Proline ($C_5H_9NO_2$, purity 99%, Merck) and L-Arginine ($C_6H_{14}N_4O_2$, purity 99 %, Merck) were added. Additionally, L-Methionine ($C_5H_{11}NO_2S$, purity 99%, Merck) at a weight percentage of 0.5, 1 and 1.5 wt.%, was used to synthesize $C/N/S/TiO_2$ catalysts. After maintaining the solutions for 24 h at room temperature, they were dried in an electric oven at 120 °C for 12 h. The samples were calcined at 500 °C for 2 h inside a muffle furnace and photocatalysts were obtained.

### 2.2. Characterization of the Photocatalysts

FESEM (Philips XL 30 and S-4160, Eindhoven, Netherlands) combined with Energy Dispersive X-rays (EDS) was used to examine the structures and sizes of nano-particles and to determine the composition of samples. It is usually recommended to coat the sample with a thin layer of gold. This will prevent the specimen from charging. It also promotes the emission of secondary electrons so that the specimen conducts evenly, and provides a homogeneous surface for analysis and imaging. A 20-kV acceleration voltage was used to obtain the EDS spectrum, and 19 s intervals were used for each measurement. The XRD analysis of the synthesized photocatalysts was carried out using a Bruker diffractometer (Karlsruhe, Germany), Cu-Kα X-rays of wavelength (λ) = 1.5406 Å, operating at 40 kV and 30 mA. An angular range of 20 to 80° was implemented to collect diffraction patterns at 25 °C. The step size was 0.05° per step, and the dwell time was 12 s per increment. The functional groups of as-prepared compounds were investigated by Fourier Transform Infrared (FTIR) spectra in KBr pellets (Shimadzu Varian 4300 spectrophotometer, Kyoto, Japan). In this case, 0.1 to 1.0% of the sample was mixed well with 200 to 250 mg of fine alkali halide powder (KBr is used as an example below) and then finely ground and placed into the pellet-forming dye. Under a vacuum of several mm Hg, approximately 8 tons of force was applied for several minutes to form transparent pellets. KBr powder was degassed to remove air and moisture. When a vacuum is not sufficient, pellets may be easily broken and scatter light as a result. The KBr powder should be pulverized to a maximum of 200 mesh before being formed into pellets. They were dried at approximately 110 °C for two to three hours. A desiccator should be used to store the powder after it has been dried. A UV-Vis spectrophotometer (UV–vis DRS) (Shimadzu 1800, Kyoto, Japan, with a resolution of 1 nm, wavelength range of 190–1100 nm, absorbance range (−4 to 4 Abs) and center height: 15 mm) was employed to evaluate the absorbance patterns of catalysts. The photoluminescence (PL, Perkin Elmer LS55, Waltham, MA, USA) emission spectra were obtained at an excitation wavelength of 410 nm, with an emission bandwidth of 5 nm. Additionally, a pulsed Xenon lamp (Juarez, Mexico) was used as a source of excitation, at 25 °C.

### 2.3. Experimental Setup and Procedure

The photocatalytic tests were conducted in a cylindrical container with a volume of 400 mL and LED lamps (50 W, emission: 405 nm, light intensity: 13 lm per $m^2$) around the inside. The reactor was made up of a double-walled quartz cylinder surrounded by fluid to keep the temperature of the solution at 25 °C. CEX or MNZ was added to purified water along with a specific amount of NaOH (1 M, Merck) and $H_2SO_4$ (1 M, Merck) to produce antibiotic solutions. After adding a specific amount of the catalyst, the solution was mixed in dark conditions for 30 min to obtain full adsorption–desorption equilibrium. Subsequently, the solutions containing suspended $C/N/TiO_2$ or $C/N/S/TiO_2$ catalysts were mixed under visible light irradiation. Then, at different times, samples were collected from the reaction solution; after separation of the catalyst, the concentration of antibiotics was measured spectrophotometrically (V-570, Jasco, Tokyo, Japan) according to the calibration curve (Figure S1a,b) at the maximum wavelength of 320 nm and 273 nm for MNZ and CEX, respectively. The removal (%) was determined as follows:

$$\text{Removal (\%)} = \left(1 - \frac{C_t}{C_0}\right) \times 100 \tag{1}$$

where $C_0$ and $C_t$ stand for the concentrations of antibiotics at time zero and t in the solution, respectively.

### 2.4. Experimental Design Methodology

To assess the effects of processing factors on response and their interactions, RSM based on the central composite design (CCD) was used to design experiments. In this study, the following four numeric factors were considered as processing factors: antibiotic concentration, catalyst concentration, pH and irradiation time, which were examined at three levels. The type of antibiotic was chosen as a categorical factor at two levels (Table 1). The total number of experiments was 60 runs (Table S1). The antibiotic degradation efficiency, its relation with the processing factors and the effects of factors were evaluated with a quadratic model and the accuracy of models was assessed with an analysis of variance (ANOVA).

**Table 1.** Experimental levels of the process parameters.

| Factors | Levels | | |
|---|---|---|---|
| | −1 | 0 | 1 |
| A: [MNZ] or [CEX] (mg/L) | 50 | 75 | 100 |
| B: [Catalyst] (g/L) | 0.5 | 1 | 1.5 |
| C: pH | 4 | 7 | 10 |
| D: Irradiation time (h) | 1 | 1.5 | 2 |
| E: Type of antibiotic | MNZ | - | CEX |

## 3. Results and Discussion

### 3.1. Optimization of Amino Acids Content in TiO$_2$ Network

The photodegradation of MNZ and CEX by TiO$_2$ with different amino acid loadings at an initial antibiotic concentration of 50 mg/L and pH of 4 is shown in Figure 1a,b. The amount of L-Proline increased from 0.5 to 2 wt.%, leading to an increase of 69% to 98% in MNZ photocatalytic degradation and 54% to 90% in CEX photocatalytic degradation after 120 min of irradiation (Figure 2a,b). In contrast, further increasing the L-Proline content to 2.5 wt.% decreased the removal efficiency of both antibiotics by approximately 2–6% and prevented further enhancement of the photocatalytic activity of the catalyst. The upward trend can be attributed to the reduction in the TiO$_2$ band gap. This occurs due to the formation of new impurity levels between TiO$_2$'s conduction band and valence band [29,30]. Nonetheless, the slight drop in degradation efficiency after increasing L-proline to 2.5 wt.% is due to the increased recombination rate of electron/hole pairs [19], which can be explained by the increasing PL intensity observed after increasing the amount of L-Proline doped on TiO$_2$ to 2.5 wt.% (Figure 2a). Moreover, the results of L-Methionine loading confirmed an improvement in the degradation efficiency as the increase in the amount of loading up to 1.5 wt.% resulted in a 99 % and 94 % removal for MNZ and CEX, respectively, after 120 min irradiation (Figure 1c,d). The high photocatalytic activity performance of L-Methionine-TiO$_2$ under visible light irradiation may be due to the introduction of C/N/S into the lattice of TiO$_2$. The doping of C and N non-metals can have positive effects on the formation of a hybrid orbital close to the valance band of TiO$_2$ by replacing O atoms with carbon or nitrogen atoms; sulfur dopants can narrow the band gap through the overlap of sulfur 3p states and oxygen 2p states. Hence, the visible photo-absorption capacity can be improved [25,31]. When the loading was increased to 2 wt.% of L-Methionine, a downward trend was observed, owing to the acceleration of the recombination of light-generated electrons and holes (Figure 2b).

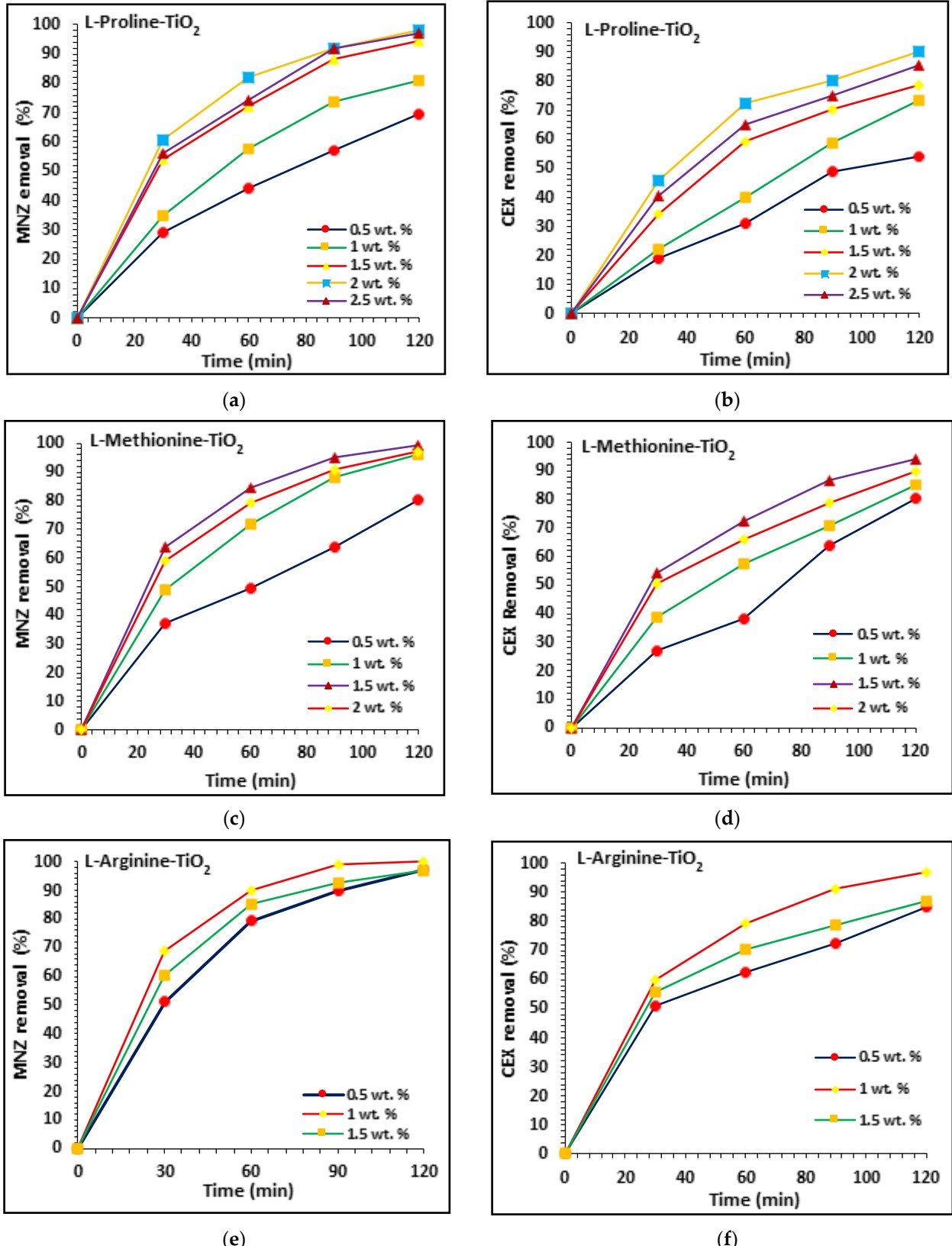

**Figure 1.** Effect of amino acid loadings on photocatalytic removal of MNZ by (**a**) L-proline-TiO$_2$ (**c**) L-Methionine (**e**) L-Arginine and CEX by (**b**) L-proline-TiO$_2$ (**d**) L-Methionine (**f**) L-Arginine under conditions of initial concentration of drugs of 50 mg/L, catalyst dosage of 1 g/L and pH of 4 (**a**,**b**).

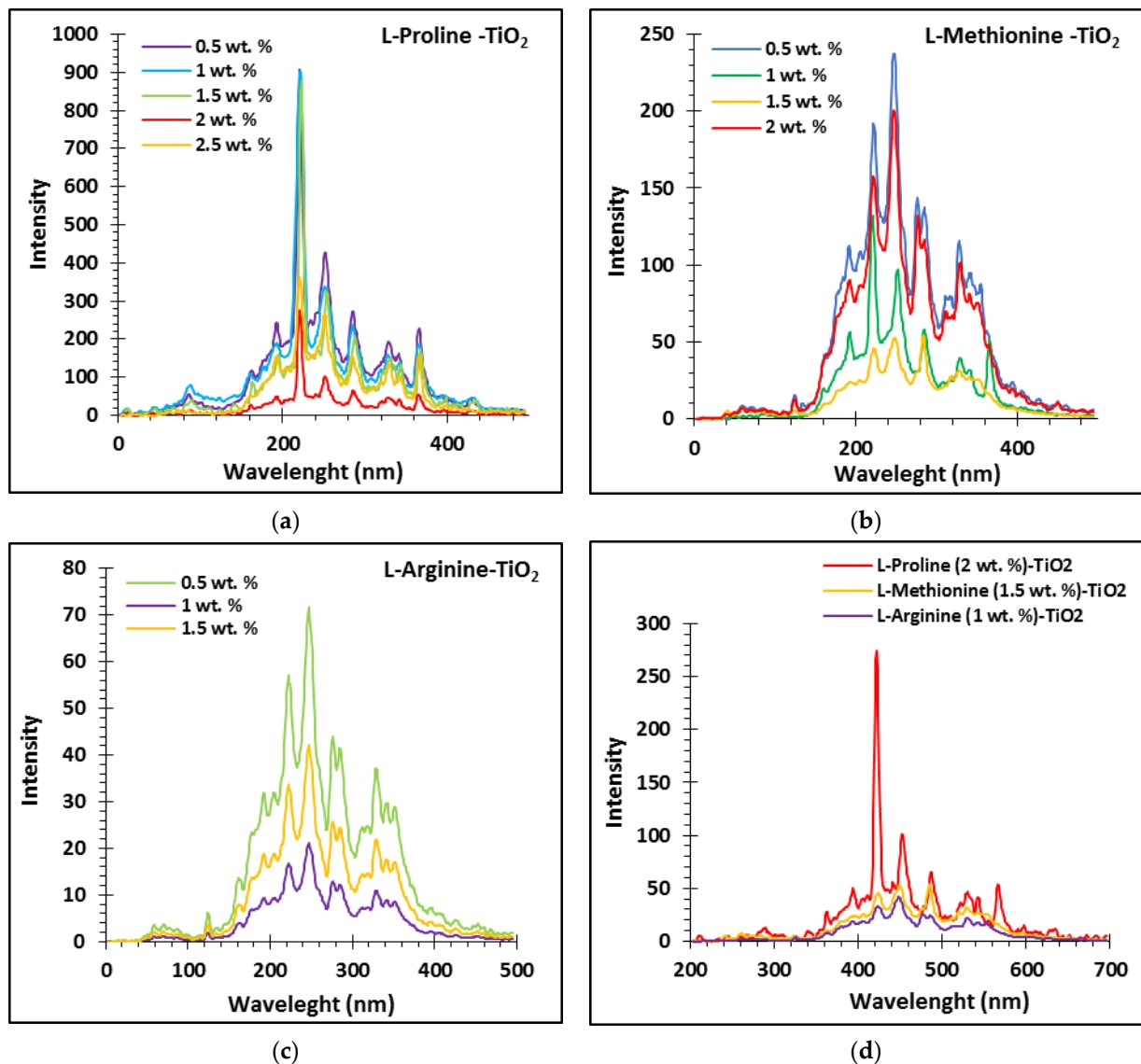

**Figure 2.** PL spectra of (**a**) TiO$_2$ with different loading of L-proline (**b**) TiO$_2$ with different loading of L-Methionine (**c**) TiO$_2$ with different loading of L-Arginine (**d**) different photocatalysts with optimum amino acid composition.

Moreover, complete removal efficiency of both antibiotics was achieved under 120 min visible light exposure by using 1 wt.% L-Arginine -TiO$_2$ as photocatalyst (Figure 1e,f). The PL spectra of L-Arginine-TiO$_2$ has the lowest intensity which proves that it possesses the lowest recombination rate of the electron/hole pairs (Figure 2c).

The kinetics of MNZ and CEX photocatalytic removal for each optimum nanocomposite (L-Proline (2 wt.%)-TiO$_2$, L-Methionine (1.5 wt.%)-TiO$_2$ and L-Arginine (1 wt.%)-TiO$_2$) are displayed in Figures S2 and S3. Table 2 shows the pseudo-first order and second order rate constants (k$_1$ and k$_2$ accordingly) and coefficient determination (R$^2$). According to the results, the pseudo-first-order model agrees satisfactorily with experimental data for all prepared nanocomposites, and its equation is as follows:

$$Ln\left(\frac{C_0}{C_t}\right) = K_1 t \tag{2}$$

$$\frac{t}{C_t} = \frac{1}{k_2 C_0} + \frac{t}{C_0} \tag{3}$$

where $C_0$, $C_t$, $k_1$, $k_2$ and t are concentrations at time zero and t, the rate constant and irradiation time, respectively. The kinetic data are fitted with a pseudo first order model for all of the photocatalysts. $k_1$ values of the pseudo first order kinetic model are about 0.0321, 0.0387 and 0.0587 $min^{-1}$ for L-Proline (2 wt.%)-$TiO_2$, L-Methionine (1.5 wt.%)-$TiO_2$ and L-Arginine (1 wt.%)-$TiO_2$, respectively. The results confirm the highest photocatalytic activity of L-Arginine (1 wt.%)-$TiO_2$, as explained previously.

**Table 2.** Pseudo first and second order models for photocatalytic removal of MNZ and CEX.

| L-Amino Acid-TiO$_2$ Photocatalysts | Weight Fraction of Amino Acid (wt.%) | Pseudo First Order Model | | Pseudo Second Order Model | |
|---|---|---|---|---|---|
| | | $k_1$ (min$^{-1}$) | $R^2$ | $k_2$ (min$^{-1}$) | $R^2$ |
| | | **MNZ** | | | |
| L-Proline-TiO$_2$ | 0.5 | 0.0096 | 0.996 | 0.0050 | 0.924 |
| | 1 | 0.0141 | 0.997 | 0.0068 | 0.904 |
| | 1.5 | 0.0239 | 0.995 | 0.0155 | 0.774 |
| | 2 | 0.0321 | 0.983 | 0.0372 | 0.64 |
| | 2.5 | 0.0291 | 0.986 | 0.0251 | 0.719 |
| L-Methionine-TiO$_2$ | 0.5 | 0.0126 | 0.974 | 0.0056 | 0.842 |
| | 1 | 0.0267 | 0.982 | 0.0193 | 0.695 |
| | 1.5 | 0.0387 | 0.991 | 0.0773 | 0.593 |
| | 2 | 0.0289 | 0.974 | 0.0264 | 0.69 |
| L-Arginine-TiO$_2$ | 0.5 | 0.0291 | 0.986 | 0.0254 | 0.671 |
| | 1 | 0.0578 | 0.997 | 0.5823 | 0.563 |
| | 1.5 | 0.0288 | 0.971 | 0.0256 | 0.671 |
| | | **CEX** | | | |
| L-Proline-TiO$_2$ | 0.5 | 0.0068 | 0.986 | 0.0047 | 0.968 |
| | 1 | 0.0109 | 0.984 | 0.0048 | 0.885 |
| | 1.5 | 0.0129 | 0.993 | 0.0065 | 0.917 |
| | 2 | 0.0188 | 0.991 | 0.01 | 0.862 |
| | 2.5 | 0.0158 | 0.995 | 0.0077 | 0.813 |
| L-Methionine-TiO$_2$ | 0.5 | 0.0131 | 0.953 | 0.0051 | 0.809 |
| | 1 | 0.0169 | 0.985 | 0.0052 | 0.801 |
| | 1.5 | 0.0235 | 0.995 | 0.0141 | 0.767 |
| | 2 | 0.0182 | 0.986 | 0.0094 | 0.789 |
| L-Arginine-TiO$_2$ | 0.5 | 0.0144 | 0.969 | 0.0074 | 0.848 |
| | 1 | 0.0161 | 0.972 | 0.0260 | 0.691 |
| | 1.5 | 0.0288 | 0.991 | 0.0091 | 0.866 |

### 3.2. Characterization of Synthesized Catalysts

Synthesized photocatalysts were investigated with SEM and EDS, as shown in Figure 3. As shown in Table S2, nano-photocatalyst particle sizes are also calculated using image processing software (ImageJ 1.44p). Thus, $TiO_2$ doped with L-Proline, L-Arginine, and L-Methionine has an average size of 37, 29 and 20 nm. Some gold nanoparticles can be observed on the surface of photocatalysts. This is because coating nanoparticles with gold increases their conductivity.

The results of EDX spectroscopy are used to identify the compounds that compose nano-photocatalysts. L-Proline-$TiO_2$ and L-Arginine-$TiO_2$ nanocomposites showed different proportions of Ti, O, C, and N elements. Figure S4 also illustrates the identical spatial composition of elements in the photocatalysts.

UV-vis diffuse reflection spectrum analysis was used to evaluate the optical properties of compounds. Figure 4 shows the UV-Vis DRS spectra and Tauc plots of the prepared photocatalysts with an optimum loading of dopants.

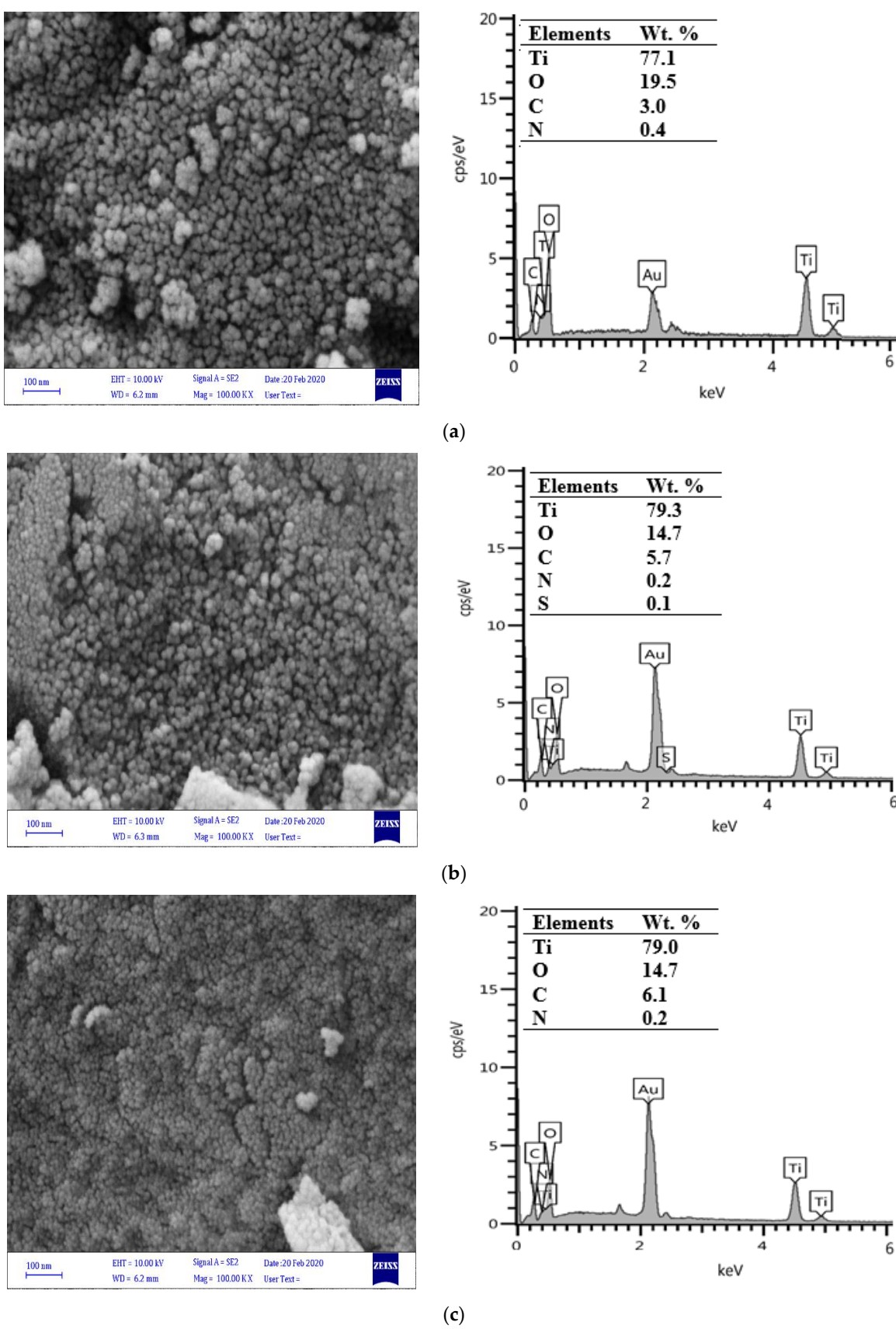

**Figure 3.** FESEM images and EDS spectra of (**a**) L-Proline (2 wt.%)-TiO$_2$ (**b**) L-Methionine (1.5 wt.%)-TiO$_2$ (**c**) L-Arginine (1 wt.%)-TiO$_2$.

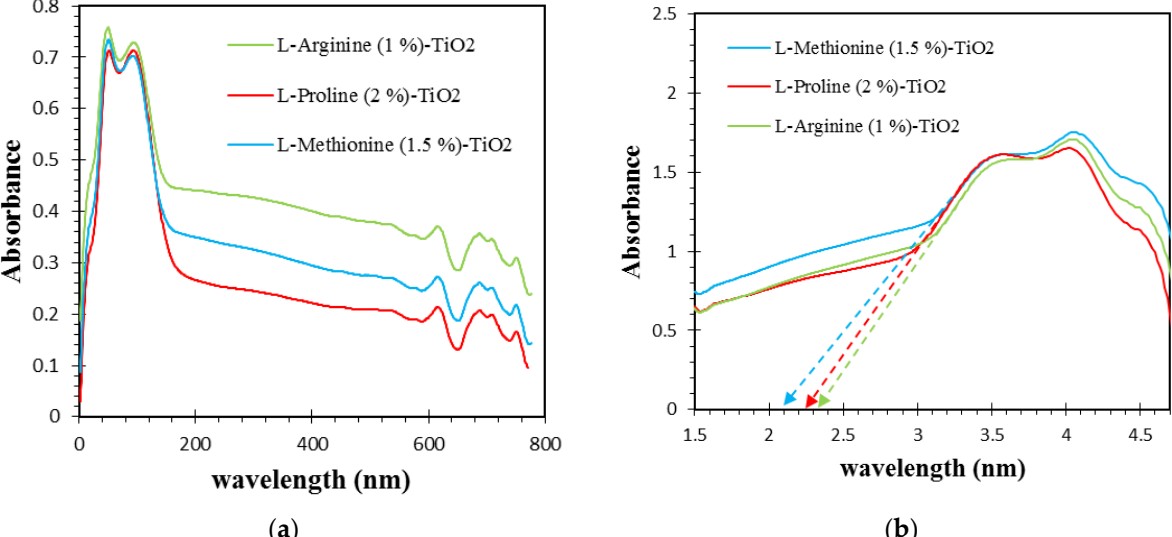

**Figure 4.** UV-vis diffuse reflectance absorption spectra (**a**) and determination of bandgap energies (**b**) of L-Proline (2 wt.%)-TiO$_2$, L-Methionine (1.5wt.%)-TiO$_2$ and L-Arginine (1wt.%)-TiO$_2$.

The modification of TiO$_2$ with amino acids led to a significant absorption extension in the visible light region. The red shift in absorption wavelength led to a wider absorption range and indicated that non-metal doping, especially co-doped or tri-doped C, N, S, had a significant effect on the photocatalytic activity of TiO$_2$ and made it applicable to the photodegradation of pollutants under visible light illumination. The Tauc plots of the catalysts displayed that the optical band gap energies of L-Arginine (1 wt.%)-TiO$_2$, L-Methionine (1.5 wt.%)-TiO$_2$ and L-Proline (2 wt.%)-TiO$_2$ are 2.3, 2.1 and 2.2 eV, proving that amino acid doping in a TiO$_2$ network plays a key role in narrowing the band gap as the band gap of pure anatase TiO$_2$ is 3.2 eV [19,32–34].

The crystallinity and phase structure of the photocatalysts are studied by performing an XRD analysis. The patterns of the nanoparticles shown in Figure 5a confirm the presence of TiO$_2$ according to JCPDS#21-1272 [17]. The XRD pattern of amino acid-TiO$_2$ in comparison to pure TiO$_2$ shows a shift in the diffraction peaks. In this case, the shift of diffraction peaks can be attributed to an increase/decrease in lattice parameters. The main reason for this effect is the difference in ionic radii between the main element and its dopant. No other peaks were detected related to carbon, nitrogen and sulfur because of their small amounts. Figure 5b displays the FTIR analysis that proved the formation of Ti-O and Ti-O-Ti bonds in the range of 400 and 800 cm$^{-1}$. The bands at 1600 and 3400 cm$^{-1}$ were assigned to OH groups in hydroxyl and adsorbed water for three catalysts. The strong peaks at 3400 cm$^{-1}$ could be attributed to NH$_2$ functional groups [19]. The vibration bonds at 2400 cm$^{-1}$ and 1200 cm$^{-1}$ could be associated with CO$_2$ and C-N groups of nanostructures, respectively.

*3.3. Model of Photodegradation of MNZ and CEX by L-Arginine (1 wt.%.)-TiO$_2$*

The analysis of variance (ANOVA) results (Table 3) of the second-order model showed that the model for antibiotics photodegradation is highly significant as its *p*-value is less than 0.0001 and its *f*-value is high (69.69) [35]. In addition, Adeq Precision indicates the signal to noise ratio (the predicted values versus experimental data) is desirable (34.29 > 4). The lack of fit is found to be not significant. The normal probability plot of the residuals (Figure S5) further demonstrated that the models fit the experimental runs accurately. Moreover, all processing parameters, including antibiotic concentration (A), catalyst dosage (B), pH (C), illumination time (D), type of antibiotic © and their interactions (AC, BC, CE) are significant parameters based on their *p*-values. The *f*-values and coefficients of Equation (4) indicate that antibiotic concentration, pH, and catalyst concentration all have

a greater effect on drug photodegradation efficiency than irradiation time or antibiotic type.

$$Y(\%) = +76.36 - 14.30 \times A + 6.26 \times B - 8.14 \times C + 4.82 \times D - 3.66 \times E + 1.68 \times A - 2.23 \times BC + 3.06 \times CE + 6.87 \times A^2 - 11.98 \times B^2 - 9.33 \times C^2 - 4.13 \times D^2 \tag{4}$$

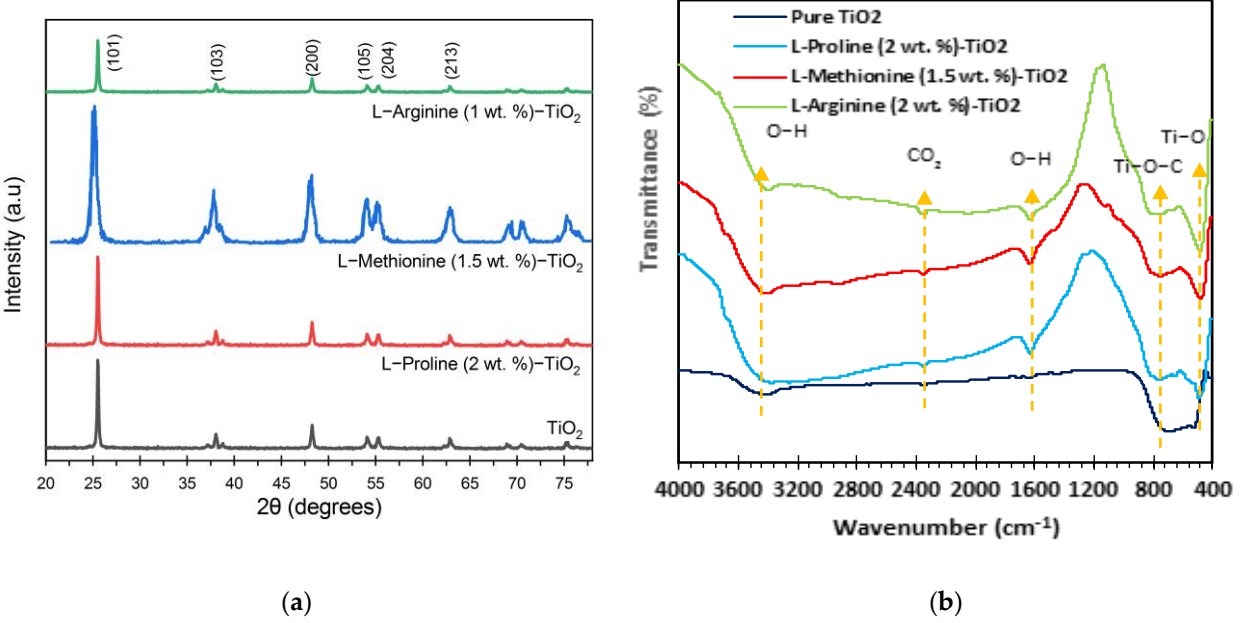

(**a**)                                   (**b**)

**Figure 5.** (**a**) XRD patterns and (**b**) FTIR spectra of the synthesized photocatalysts.

**Table 3.** ANOVA results for the model.

| Source | Mean Square | df | $f$-Value | $p$-Value |
|---|---|---|---|---|
| Model | 1512.32 | 12 | 69.69 | <0.0001 |
| A: [MNZ] or [CEX] | 7361.64 | 1 | 339.26 | <0.0001 |
| $B_2$: [Catalyst] | 1408.75 | 1 | 64.92 | <0.0001 |
| C: pH | 2384.69 | 1 | 109.90 | <0.0001 |
| D: Irradiation time | 835.21 | 1 | 38.49 | <0.0001 |
| E: Type of antibiotic | 803.74 | 1 | 37.04 | <0.0001 |
| AC | 90.79 | 1 | 4.18 | 0.0464 |
| BC | 158.87 | 1 | 7.32 | 0.0095 |
| CE | 337.33 | 1 | 15.55 | 0.0003 |
| $A^2$ | 244.52 | 1 | 11.27 | 0.0016 |
| $B^2$ | 743.78 | 1 | 34.28 | <0.0001 |
| $C^2$ | 451.14 | 1 | 20.79 | <0.0001 |
| $D^2$ | 88.42 | 1 | 4.07 | 0.0493 |
| Residual | 21.70 | 47 | - | - |
| $R^2$ = 0.95 | $R^2_{adj}$ = 0.93 | Lack of fit = 0.25 | C.V. = 7.14 | Adeq Precision = 34.29 |

The results of the effect of processing factors on photodegradation of MNZ and CEX by L-Arginine (1 wt.%.)-TiO$_2$ are shown in Figures 6 and 7.

According to the counter plot of antibiotic concentration versus irradiation time (Figure 6a,b), increasing the pollutant concentration from 50 to 100 mg/L adversely affected degradation efficiency, which declined to 62% and 50%, respectively, for MNZ and CEX. This trend can be attributed to the saturation of the active sites of the photocatalysts. In addition, high concentrations of antibiotics reduce the number of photons that reach the surface of photocatalyst particles. In turn, this can negatively impact hydroxyl radical generation and other reactive species [36].

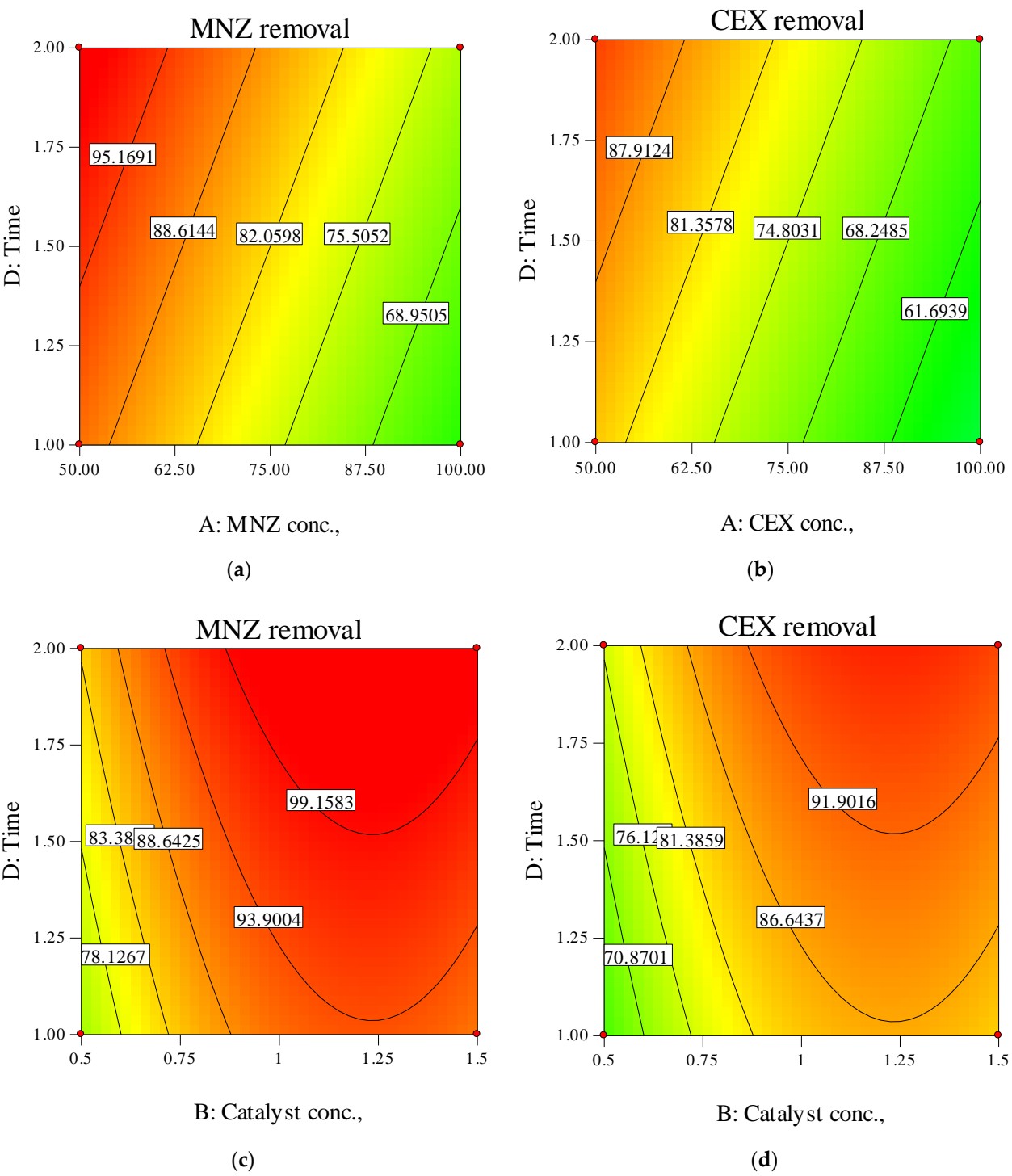

**Figure 6.** Counter plot of MNZ (**a**) and CEX (**b**) removal in terms of antibiotic concentration and irradiation time (at pH of 4 and catalyst loading of 1.5 g/L) and MNZ (**c**) and CEX (**d**) removal as a function L-Arginine (1 wt.%)-TiO$_2$ dosage and irradiation time (at pH of 4 and drug concentration of 50 mg/L).

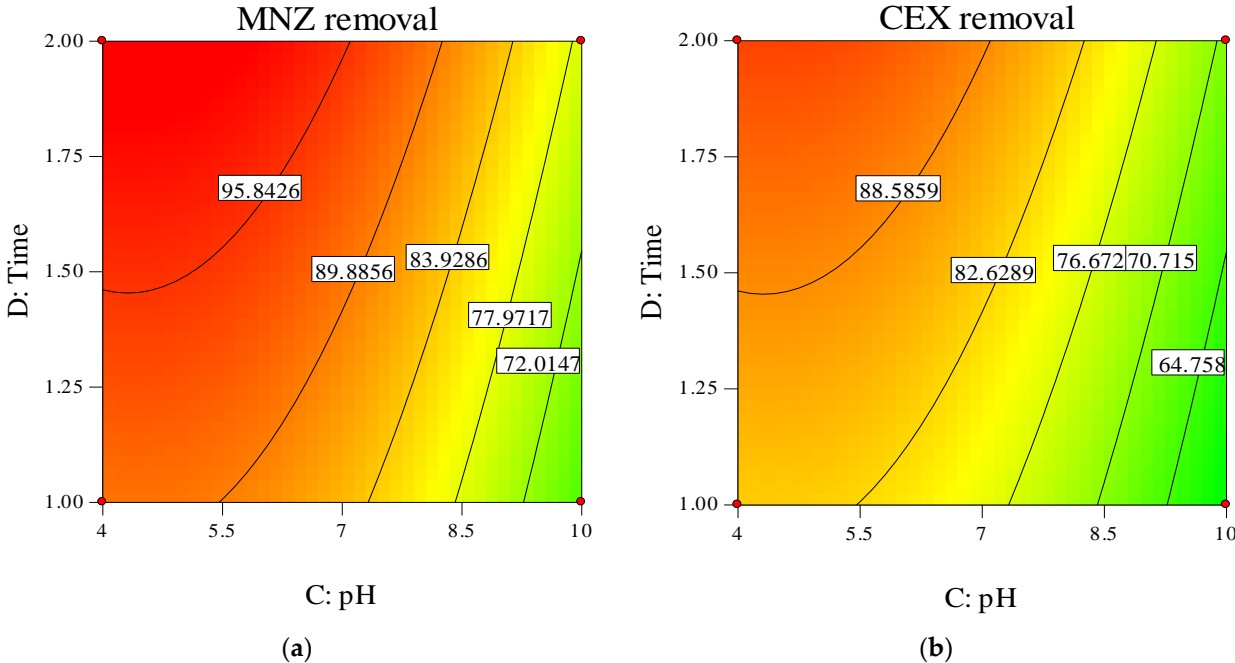

**Figure 7.** Counter plot of MNZ (**a**) and CEX (**b**) removal in terms of pH and irradiation time at conditions of drug concentration of 50 mg/L and catalyst dosage of 1.5 g/L.

The photodegradation efficiency of both antibiotics experienced an increase to more than 90% following an increase in the photocatalyst concentration of up to 1.5 g/L at an acidic pH after 120 min irradiation time (Figure 6c,d) as a result of a greater generation of reactive species and the existence of more active sites [33].

The effects of pH and irradiation time on the response are illustrated in Figure 7. When the irradiation time increased from 1 to 2 h, a 20–30% increase in antibiotic removal efficiency was observed. The longer exposure time increases the time for radical species to react with pollutants, and generates more reactive species that enhance photocatalytic activity. Based on the evaluation of the effects of pH, higher MNZ and CEX photodegradation was observed in acidic media. The catalyst surface is negatively charged above pH$_{PZC}$ and positively charged below pH$_{PZC}$. The pH$_{PZC}$ of L-Arginine (1 wt.%)-TiO$_2$ was approximately 6.4 (Figure S6). MNZ with pKa = 2.55 [37] appears in an anionic form at a pH of more than 2.55. The electrostatic attraction at pH = 4 between the surface of the catalyst and MNZ leads to a higher photocatalytic performance of L-Arginine (1 wt.%)-TiO$_2$ nanoparticles for MNZ removal at an acidic pH. Moreover, CEX with pKa values of 2.6 and 6.9 for the carboxyl group and amine group, respectively, has a positive charge at pH < 2.6 and it has a negative charge at pH > 6.9; however, it can be observed in neutralized form at a pH in the range of 2.6 to 6.9 [38].

The optimum conditions for the photodegradation of MNZ based on the desirability of 1 were identified at an MNZ concentration of 50 mg/L, pH = 4, L-Arginine (1 wt.%)-TiO$_2$ catalyst concentration of 1.5 g/L under 90 min irradiation time. MNZ photodegradation efficiency was obtained at 99.9% under these conditions, which was in agreement with the predicted removal efficiency (100%), confirming the validity of the model. Moreover, 97.2% of CEX was removed under the optimum conditions of a 50 mg/L CEX concentration, 1 g/L catalyst concentration, at pH 4 and after 120 min irradiation. The reusability of L-Arginine-TiO$_2$ catalyst was investigated in optimum conditions for the photodegradation of MNZ and CEX; it was found to have high photocatalytic performance even after five cycles, incurring a photodegradation loss of only about 7–10% (Figure 8).

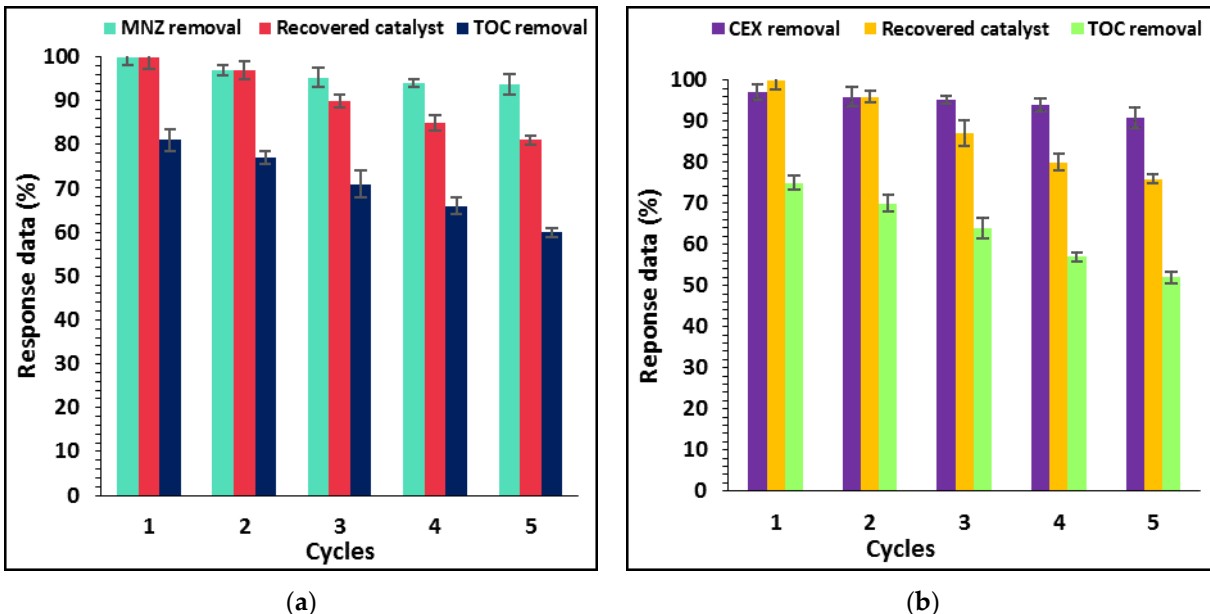

**Figure 8.** The results of reusability of the photocatalyst for degradation of (**a**) MNZ and (**b**) CEX.

*3.4. Possible Photodegradation Mechanism*

In the valence band (VB) and conduction band (CB) of the photocatalyst, these photo-generated electrons and holes interact with $H_2O$ and $O_2$. The identification of the primary active species is crucial to proposing a photocatalytic mechanism. Consequently, a number of scavengers were introduced to the reaction solution in order to perform radical trapping experiments. In this study, ammonium oxalate (AO), carbon tetrachloride (CT), isopropanol alcohol (IPA) and sodium azide (SA) as scavengers of $h^+$, superoxide radicals ($O_2^{\bullet-}$), hydroxyl radicals ($^\bullet OH$), and singlet oxygen ($^1O_2$), respectively, were introduced to the solution under optimum conditions. (Figure 9). MNZ and CEX photocatalytic removal efficiencies were reduced with the addition of scavengers. The introduction of IPA significantly reduced photocatalytic efficiency, which shows that $^\bullet OH$ is the main active species in the MNZ and CEX photodegradation process.

The UV-Vis DRS analysis proved that L-Arginine (1 wt.%) is a visible driven nanophotocatalyst with a band gap energy of 2.3 eV. Therefore, under visible light irradiation, photoinduced electron and holes are generated, resulting in the formation of active radical species for the decomposition of MNZ or CEX drugs, as shown in the Figure 10. As a result of C and N co-doping, the band gap of L-Arginine-$TiO_2$ can be effectively excited by visible light by increasing the VB energy from O2p to N2p. The reaction between dissolved oxygen ($O_2$) and photogenerated electrons produces reactive species ($O_2^{\bullet-}$, $HO_2^\bullet$ and $^\bullet OH$), resulting in the degradation of MNZ and CEX molecules [39]. In addition, photoinduced holes and adsorbed MNZ and CEX on the photocatalyst interface can directly generate reactive $MNZ^+$ and $CEX^+$ radicals.

*3.5. Effect of Inorganic Anions and Humic Acid on Photocatalysis Process*

The influence of common coexisting anions in water resources (sulfate ($SO_4^{2-}$), bicarbonate ($HCO_3^-$), nitrate ($NO_3^-$) and chloride ($Cl^-$)) and humic acid (HA) on photocatalysis was also evaluated, as shown in Figure 11. As observed, all anions show a slight inhibitor effect on the MNZ and CEX photodegradation efficiency. The highest reduction in MNZ and CEX removal efficiencies was observed by introduction of bicarbonate anions because it reacts to $^\bullet OH$ and $h^+$ and play the role of scavengers [40]. The removal efficiencies of MNZ and CEX decrease significantly with the addition of 30 mg/L of HA (Figure 11). Due to the adsorption of HA molecules on the surface of the nanophotocatalyst, the active sites become saturated, reducing the formation of hydroxyl radicals and other reactive species.

Furthermore, HA molecules are powerful free radical scavengers, especially when it comes to •OH [19].

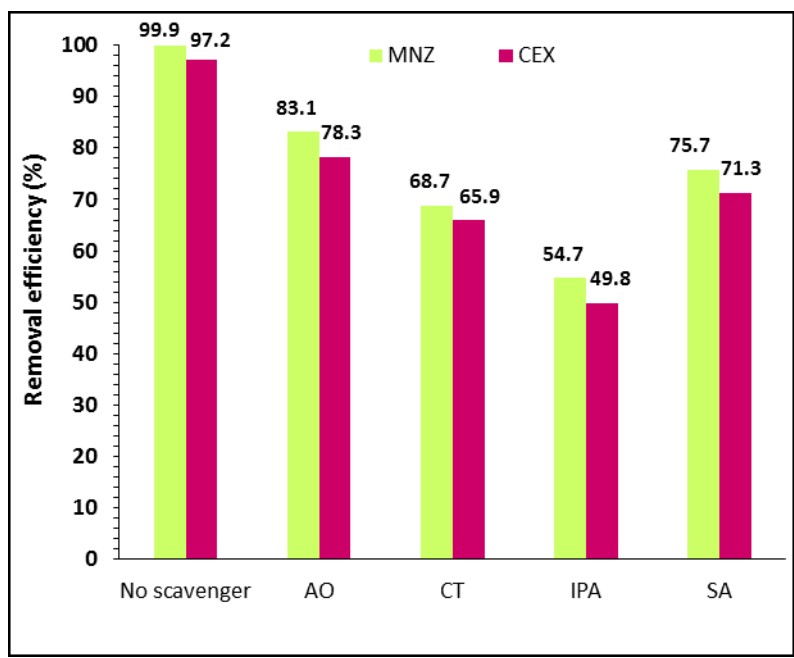

**Figure 9.** Effect of scavengers on MNZ and CEX photodegradation at optimum conditions (pH = 4, [drug] = 50 mg/L and catalyst concentration of 1.5 g/L after 120 min).

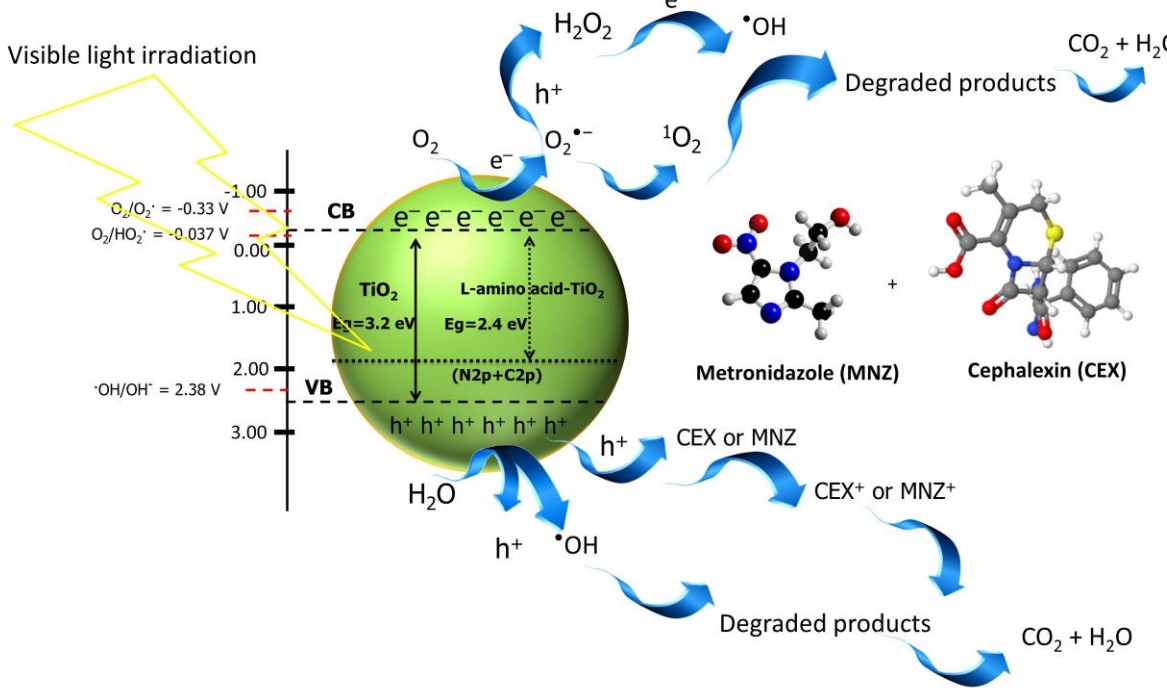

**Figure 10.** Schematic diagram of photodegradation of MNZ and CEX over L-Amino acid-$TiO_2$.

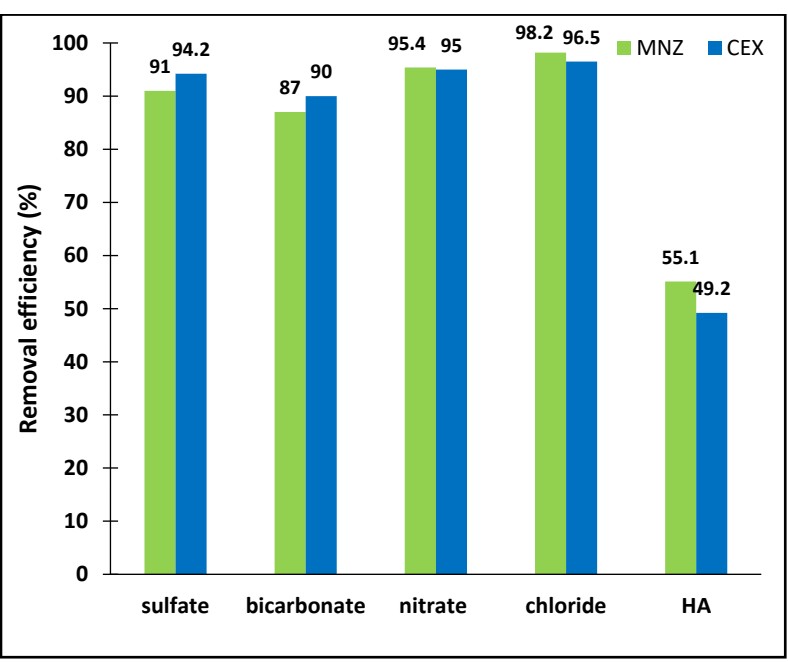

**Figure 11.** Effect of competing water anions (100 mg/L) and HA (25 mg/L) during MNZ and CEX photodegradation in optimum conditions (pH = 4, [drug] = 50 mg/L and catalyst concentration of 1.5 g/L after 120 min).

## 4. Conclusions

An evaluation of the results of the photodegradation of antibiotics using co-doped and tri-doped $TiO_2$ confirms that the modification of $TiO_2$ with C, N and S plays a vital role in improving the photocatalytic properties of the catalyst, such as by making the catalyst active in the visible light region and reducing the recombination rate of $e^-$ and $h^+$ pairs. A higher photocatalytic efficiency was observed for MNZ removal than CEX using L-Arginine (1 wt.%)-$TiO_2$ nanostructures under visible light illumination. The statistical results based on RSM proved the validity and adequacy of the model with a high F-value and $R^2$. Furthermore, the ANOVA results showed that the effects of antibiotic concentration, pH and catalyst concentration on the photodegradation efficiency were more significant than two other factors (irradiation time and antibiotic type). The optimal conditions were 50 mg/L, pH 4, and 1.5 g/L L-Arginine (1 wt.%)-$TiO_2$ catalyst concentration under 90 min irradiation time to achieve 99.9% photodegradation efficiency for MNZ. CEX removal of 97.2% was also achieved in the optimum conditions of 50 mg/L CEX, 1g/L catalyst, pH 4 and 120 min of irradiation. The catalyst showed high reusability without significant loss in its photocatalytic performance for MNZ and CEX removal even after five cycles. The result of the trapping experiment shows that $^\bullet OH$ are the main active species in the removal of MNZ and CEX. The high photocatalytic performance and stability of L-arginine (1 wt.%) make it a promising photocatalyst for practical and industrial applications.

**Supplementary Materials:** The following supporting information can be downloaded at: https://www.mdpi.com/article/10.3390/w15030535/s1, Figure S1: Calibration curve (a) MNZ, (b) CEX; Figure S2: Kinetic models of MNZ photocatalytic removal for L-Amino acid-$TiO_2$ at catalyst loading of 1 g/L, MNZ concentration of 50 mg/L and pH of 4.; Figure S3: Kinetic models of CEX photocatalytic removal for L-Amino acid-$TiO_2$ at catalyst loading of 1 g/L, CEX concentration of 50 mg/L and pH of 4; Figure S4: Elemental mapping of the prepared nano-photocatalysts; Figure S5: Predicted vs. actual values plot for antibiotics photodegradation process; Figure S6: Measurement of the zero charge point for L-Arginine-$TiO_2$ photocatalysts; Table S1: Experimental conditions and response data for drug photocatalytic removal; Table S2: The particle size distribution of the prepared samples.

**Author Contributions:** Conceptualization, H.Z., S.A.M., P.E. and E.A.; methodology, H.Z., P.E. and E.A.; software, H.Z., P.E. and E.A.; formal analysis, H.Z., P.E., E.A., J.F. and S.M.; investigation, H.Z., J.F. and S.M.; resources, S.A.M.; data curation, H.Z., S.A.M., P.E., E.A., J.F. and S.M.; writing—original draft preparation, H.Z. and S.A.M.; writing—review and editing, E.A.; visualization, H.Z.; supervision, H.Z. and S.A.M.; project administration, H.Z., S.A.M., J.F. and S.M.; funding acquisition, S.A.M. All authors have read and agreed to the published version of the manuscript.

**Funding:** This research was funded by Kermanshah University of Medical Sciences grant number 97649 and ethical code: IR.KUMS.REC.1397.587.

**Data Availability Statement:** Data are contained within the article.

**Acknowledgments:** The authors are grateful to Kermanshah University of Medical Sciences.

**Conflicts of Interest:** The authors declare no conflict of interest.

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
