# Peer review of "Comparative Study on Photocatalytic Performance of TiO2 Doped with Different Amino Acids in Degradation of Antibiotics"

_water, doi:10.3390/w15030535_

Round 1

Reviewer 1 Report

Review of the manuscript Water-2117302 for the Authors: This study presents a kinetic study of amino acid doped TiO2 composite assisted photocatalytic degradation of antibiotics CEX and MNZ. The results presented are scientifically explained moderately well and the experimental work is robust, however the manuscript must be heavily revised in terms of language. Namely, there are many spelling mistakes as well as grammatically incorrect sentences, both in text and in figures. I suggest for now major revision.

Title – Incorrectly written and in my opinion too long. “Comparing photocatalytic activity of L-Proline-TiO2, L-Arginine-TiO2 and L-Methionine-TiO2 for antibiotic degradation from water and wastewater: A kinetic study”

Abstract – Could be more clearly presented.

Introduction – I would advise to separate the references so that the reader can know which reference corresponds to which part of the sentence, you have mostly referenced your text with stacked references.

Materials and methods – In my opinion the most scientifically lacking. The point of publishing papers is to widespread knowledge on how to reproduce or advance certain experiments. To do that, one would need to be able to compare the results they obtain by their characterization methods, so I would highly recommend you expand the characterization section by providing more detail, e.g. which retention time and step was used to obtain diffraction patterns, what were the scan rates for each of the spectroscopy techniques used. Also, refrain from using jargon-like phrases, such as “charge carrier situation”. Additionally, for FESEM analysis, state if you coated your samples with gold or not, because EDS shows traces of gold on your samples.

Results – Regarding the section of optimization, where you mention PL intensities and charge carrier recombination, there are a few mistakes that need to be corrected. First of all, PL emission spectra can only show you charge recombination, while it tells you nothing about exciton separation nor charge generation (the latter you can obtain by obtaining proper absorption spectra). Furthermore, if you were to support the numerous claims in your discussion related to time dependent charge recombination, e.g. rates and acceleration, you would need to do time transient PL spectroscopy. In diffraction analysis, try putting a pure p25 TiO2 diffractogram for comparison, obviously your 2 theta shift is not the same for all of the samples so it would be interesting to see and discuss. I would strongly suggest you revise your DRS results and method of optical bandgap determination (Tauc plots) because considering the appearance of the spectra, the extrapolation seems extremely suspicious, resulting in optical bandgap values too low to be considered true. On the contrary, cite or present literature that contains amino acid (or similar compound) doped TiO2 of similar bandgap values. Regarding FESEM analysis, please do highlight that the very small particle like morphology on your nanomaterials are indeed Au NPs. Also for the FTIR please explain what kind of peak/band do you see at 800 cm-1??

Conclusions – Clearly sum up the most important results, highlighting important values and numbers.

Literature – Should be clearly cited, so that the reader can correspond each sentence to a reference.

Author Response

The Editor of water journal

Special Issue: Heterogeneous Catalytic Processes and Advanced Nanostructures for Remediation of Contaminated Water

Subject: Revised paper

With reference to your email dated 20 December 2022, I wish to accept my deepest thanks for the reviewers' comments on the captioned paper, “Comparison the photocatalytic activity of L-Proline-TiO2, L-Arginine-TiO2 and L-Methionine-TiO2 for antibiotic degradation from water
and wastewater: Kinetic study”. I have revised the paper based on the reviewer’s greatly helpful comments and suggestions. The list of changes or the rebuttals against each point raised by the respectful editor and reviewers and the revised manuscript are submitted herewith. Your kind consideration regarding this matter is very much appreciated.

Thank you

Yours Sincerely,

Seyyed Alireza Mousavi (Corresponding Author)

Department of Environmental Health Engineering, School of Public Health, and Research Center for Environmental Determinants of Health (RCEDH), Health Institute, Kermanshah University of Medical Sciences, Kermanshah, Iran.

 Social Development and Health Promotion Research Center, Kermanshah University of Medical Sciences, Kermanshah, Iran.

Reviewer 2 Report

The manuscript is well organized and written. The study was well designed and the conclusions are consistent with the results. The paper addresses an interesting subject and presents an important set of data. However, some shortcomings must be addressed and clarified:

The manuscript is too long and has an excessive number of figures. The authors should consider the possibility of placing some of those figures as supplementary material.

The title should be revised to reflect the study carried out. The authors studied the photocatalytic activity of L-Proline-TiO2, L-Ar-1 Aginine-TiO2 and L-Methionine-TiO2 for antibiotic degradation only in water solutions and not in wastewater.

Lines 77, please define de abbreviation RSM.

Figures 1 and 2: Please identify a), b), c) ...

In the conclusions, please comment on the feasibility of using this technology at a field scale and for real wastewater.

Author Response

(The authors gave the same response as above.)

Round 2

Reviewer 1 Report

The authors have improved quite a lot, bit still some issues are present, so my recommendation is minor review.

First of all the language and style still require upgrades, secondly, the XRD pattern of pure TiO2 is not added in Figure 5a, please correct this.
